# Diagnostic performance of four lateral flow immunoassays for COVID-19 antibodies in Peruvian population

**Rodrigo Calderon-Flores**[1], **Guillermo Caceres-Cardenas**[1], **Karla Alí**[1], **Margaretha De Vos**[2], **Devy Emperador**[2], **Tatiana Cáceres**[1], **Anika Eca**[1], **Luz Villa-Castillo**[1], **Audrey Albertini**[2], **Jilian A. Sacks**[2], **Cesar Ugarte-Gil**[1,3] *

1 Instituto de Medicina Tropical Alexander von Humboldt, Universidad Peruana Cayetano Heredia, Lima, Perú, 2 FIND, the Global Alliance for Diagnostics, Geneva, Switzerland, 3 School of Medicine, Universidad Peruana Cayetano Heredia, Lima, Perú

* cesar.ugarte@upch.pe

**Data Availability Statement:** The data that support the findings of this study are openly available in

## Abstract

Serological assays have been used in seroprevalence studies to inform the dynamics of COVID-19. Lateral flow immunoassay (LFIA) tests are a very practical technology to use for this objective; however, one of their challenges may be variable diagnostic performance. Given the numerous available LFIA tests, evaluation of their accuracy is critical before real-world implementation. We performed a retrospective diagnostic evaluation study to independently determine the diagnostic accuracy of 4 different antibody-detection LFIA tests: Now Check (Bionote), CareStart (Access bio), Covid-19 BSS (Biosynex) and OnSite (CTK Biotech). The sample panel was comprised of specimens collected and stored in biobanks; specifically, specimens that were RT-PCR positive for SARS-CoV-2 collected at various times throughout the COVID-19 disease course and those that were collected before the pandemic, during 2018 or earlier, from individuals with upper respiratory symptoms but were negative for tuberculosis. Clinical performance (sensitivity and specificity) was analyzed overall, and subset across individual antibody isotypes, and days from symptoms onset. A very high specificity (98% - 100%) was found for all four tests. Overall sensitivity was variable, ranging from 29% [95% CI: 21%-39%] to 64% [95% CI: 54%-73%]. When considering detection of IgM only, the highest sensitivity was 42% [95% CI: 32%-52%], compared to 57% [95% CI: 47%-66%] for IgG only. When the analysis was restricted to at least 15 days since symptom onset, across any isotype, the sensitivity reached 90% for all four brands. All four LFIA tests proved effective for identifying COVID-19 antibodies when two conditions were met: 1) at least 15 days have elapsed since symptom onset and 2) a sample is considered positive when either IgM or IgG is present. With these considerations, the use of this assays could help in seroprevalence studies or further exploration of its potential uses.

Figshare repository at https://doi.org/10.6084/m9.figshare.21801280.v1.

**Funding:** The independent performance evaluation studies for COVID-19 diagnostic tests were funded as part of FIND's work as co-convener of the diagnostics pillar of the Access to COVID-19 Tools (ACT) Accelerator.

**Competing interests:** The authors have declared that no competing interests exist.

## Background

Since December 2019, the COVID-19 pandemic, caused by the severe acute respiratory syndrome coronavirus 2 (SARS-CoV-2), has had a devastating impact on the world population, killing and infecting more than 6.5 and 625 million people as of October 2022 [1]. To stop the spread of this pandemic, there are multiple types of COVID-19 tests which have various intended uses. Briefly, reverse transcription polymerase chain reaction (RT-PCR), and other nucleic acid amplification tests, have been shown to have the highest sensitivity and specificity for diagnosis, but can be costly [2]; for lower-cost and more rapid diagnosis, antigen rapid diagnostic tests (AgRDT) have been utilized [3]; as well, assays that detect SARS-CoV-2-specific antibodies, i.e. serological testing, may be considered for research, monitoring or diagnostic purposes [4].

After acquiring SARS-CoV-2 infection, a person normally develops a humoral immune response including the production of antibodies against certain viral antigens such as the nucleocapsid (N) protein and the spike (S) protein [5, 6]. On average, IgM and IgG antibodies against viral proteins (N and S) can be detected in serum samples after the first week from symptom onset, although this can vary depending on the host and test characteristics [7, 8]. Similarly, when deciding between RT-PCR and AgRDTs, there are some aspects to consider when choosing an antibody test.

Currently, there are four methods for antibody detection: lateral flow immunoassay (LFIA), chemiluminescence immunoassays (CLIA), enzyme-linked immunosorbent type assays (ELISA) [9], and antibody neutralization test [4]. Of these, the LFIA is the fastest, with the lowest cost and simplest method to detect antibodies; however, this methodology has been shown to have the lowest sensitivity [10]. Of note, neutralization assays have been recognized as a proxy for protective immunity to SARS-CoV-2 [11, 12]. Thus, though LFIA antibody detection may enable rapid reporting of previous virus exposure (through infection or vaccination), these tests have presented different challenges, including inappropriate use cases [13], underscoring the importance of recognizing the strengths and limitations of antibody testing using this test format to inform their optimal use.

During the early stages of the pandemic, countries such as Peru [14–17], Puerto Rico, Venezuela, and Ecuador, implemented the antibody testing to detect active COVID-19 cases [18], as molecular testing was not readily available due to critical logistic limitations. Unfortunately, antibody expression is limited during the first week of infection, thereby the window of opportunity to efficiently detect infection, isolate and stop transmission chains was lost. In addition, these tests went under minimal to no external validation, affecting the interpretation of their results [13]. Over time, countries steadily increased the use of direct virological detection methods using molecular testing and AgRDTs for diagnosis, and used antibody testing for other purposes.

There are roles -albeit with some limitations- for the use of antibody testing. Currently, these tests are commonly used as a tool for seroprevalence studies [19–21], contributing to the understanding of the immune response to the virus [22]. Even with the advent of vaccines, depending on the vaccine platform used, antibody testing can also be used to distinguish between vaccinated and non-vaccinated individuals [23–25]. This is because vaccines generate an immune response towards a specific protein (e.g. the S subunit antigen of the virus); as a consequence, we can use tests that detect antibodies against S subunits to evaluate vaccination status and tests that detect antibodies against N subunit to evaluate past or recent infection status. Additionally, a series of studies have evaluated the association between quantitative antibody tests and neutralizing antibody tests to determine if the former tests could also have a role in detecting protective immunity [26–29]. More innovatively, a study has evaluated an

LFIA antibody test coupled with a spectrum-based reader for neutralizing antibodies detection and found a high correlation with a surrogate neutralization test [30, 31].

As there is still room for use and further research on antibody testing, every manufacturer must make a thorough evaluation of its test, and this should be followed by independent validations of the test. In this study, we evaluated the accuracy of four LFIA antibody tests by using pre-pandemic samples and PCR-confirmed COVID-19 samples. Our results contribute to the validation of these tests brands to be used in seroprevalence studies and, with this validation, these brands could be tested in the future for other potential uses that will require exploring their correlation with protective immunity or ability to detect vaccination/past infection status.

## Methods

### Study design

This was a retrospective diagnostic evaluation study to independently determine the accuracy of 4 different LFIA antibody rapid tests (RTs) using de-identified samples from a biobank collection. The 4 included tests were: Now Check IgM/IgG (Bionote) [32], CareStart IgM/IgG (Access Bio) [33], Covid-19 BSS IgG/IgM (Biosynex SA) [34], and OnSite IgG/ IgM (CTK Biotech, Inc) [35].

### Study setting

The study was conducted in February 2021 at the Instituto de Medicina Tropical Alexander von Humboldt (IMTAvH) from the Universidad Peruana Cayetano Heredia (UPCH) in Lima, Peru.

### Study samples and sample size

Samples came from biobank collections. 100 SARS-CoV-2 Positive serum specimens were used from a FIND biobank collection, located at UPCH. Collected between August 2020 and June 2021, these samples come from participants who were confirmed to be infected with SARS-CoV-2 by documentation of a patient-matched positive RT-PCR test using a respiratory specimen such as an oropharyngeal (OP) swab. Samples were further categorized into subgroups according to days from symptom onset (d.f.s.o): samples collected within 0–7 days, 8–14 days, and 15+ days from symptom onset.

100 negative samples were obtained from stored sample banks collected through tuberculosis studies carried out by FIND with IMTAvH over the last 5 years, from individuals who presented with upper respiratory symptoms but tested negative for tuberculosis. Samples were obtained before November 2019, prior to the introduction of SARS-CoV-2 and thus are from participants who are not expected to have had any exposure to SARS-CoV-2.

The sample size was determined to have reasonable confidence and precision to estimate the performance of each index test for the detection of SARS-CoV-2-specific antibodies. Using 100 RT-PCR positives and 100 COVID-negative samples would yield the following confidence/precision to describe performance: a sensitivity/specificity estimate of 99%; 95%; 90%; 85%; and 75%, would yield a 95% confidence interval of +- 3%; 5%; 6%; 7%; and 8,5%, respectively.

### Study procedures

All personnel were previously trained in the use of the kits, following the manufacturer supplied instructions for use. The main objective was to evaluate the sensitivity and clinical

specificity of the COVID-19 antibody LFIA tests and determine the association of index test sensitivity by day from symptom onset.

In order to determine the clinical performance of the test, we established two categories for comparison:

- COVID-19 positive: using plasma from a patient-matched positive RT-PCR test using a respiratory specimen such as an OP swab.

- COVID-19 negative: using negative plasma samples obtained in 2018 or earlier.

### Statistical analysis

Index test clinical performance was calculated in comparison to the sample's COVID-19 positivity or negativity status (according to cohort). Analysis was performed for individual antibody isotypes and overall antibody positivity (if either IgM or IgG was positive), if applicable.

Sensitivity was defined as = [TP / (TP + FN)] x 100, where: TP (true positive) is the number of positive index test results in agreement with COVID-19 positivity, and FN (false negative) is the number of negative index test results discordant with COVID-19 positivity.

Specificity was defined as = [TN / (TN + FP)] x100, where TN (true negative) is the number of negative index test results in agreement with COVID-19 negativity, and FP (false positive) is number of positive index test results discordant with COVID-19 negativity.

The 95% confidence intervals were calculated to assess the level of uncertainty induced by sample size, using the Wilson's score method.

### Regulatory and ethics considerations

An IRB approval was obtained prior to the execution of the study (UPCH IRB SIDISI: 202569). The study also was registered in the Peruvian COVID-19 study database PRISA (EI00000001341). The use of an informed consent was not necessary since all archived samples were collected from individuals who provided informed consent. All archived samples were de-identified and participant confidentiality was maintained.

The study was conducted in accordance with ethical principles derived from international guidelines including the Declaration of Helsinki, Good Clinical Practice Guidelines: ICH GCP E6 (R2) and local laws and regulations. The results from the tests under evaluation were used only for research purposes.

## Results

A total of 200 samples were evaluated using the four antibody tests brands. Of these 200 samples, 100 of them were PCR positive for COVID-19 and the remaining 100 were from individuals who were unlikely to have had any exposure to SARS-CoV-2, these samples were obtained prior to November 2019. Of note, one PCR positive sample was excluded from the analyses because the date of symptom onset was not available. Table 1 shows the baseline characteristics of the participants whose samples were used in the study. There is relative homogeneity between both groups regarding sex, age, and presence of respiratory symptoms. The predominant upper respiratory symptom for both groups was the presence of cough (89% and 100%). The majority of the positive samples (50%) were collected within the participant's first week of symptoms.

The sensitivity and specificity estimates for each test evaluated, overall and separated by antibody isotype (IgM, IgG, and IgM or IgG) and days of symptoms, are presented in Fig 1,

**Table 1. Basal characteristics of the participants whose samples were used for the analyses.**

| Characteristics | | RT-PCR Positive | Pre-pandemic samples |
|---|---|---|---|
| | | **N = 99** | **N = 100** |
| Female (%) | | 47 (47%) | 40 (40%)* |
| Median age in years (IQR) | | 35 (31–50) | 31 (24–45)* |
| Respiratory symptoms (%) | | 89 (89%) | 100 (100%) |
| Days of symptoms | | | |
| | 0–7 days | 51 (50%) | NA |
| | 8–14 days | 38 (38%) | NA |
| | 15 + days | 12 (12%) | NA |

*These calculations are based on 98 RT-PCR Negative participants, the remaining 2 participants had missing data for gender and age.

NA: Non-applicable, these patients were suspected TB patients and they had more than 14 days of symptoms.

IQR: Interquartile range

Respiratory symptoms are considered any of the following: Cough, dyspnea, sore throat, fever, nasal congestion, or sputum production.

sorted in descending order according to sensitivity. Across the entire specimen panel, the sensitivity ranged from 64% [95% CI: 54%-73%] to 29% [95% CI: 21%-39%]. However, the sensitivity of the tests varies when subset by days from symptom initiation as well as by isotype. The highest sensitivity is noted at 94% [95% CI: 84%-98%] for samples from subjects with 15+ days from symptom initiation, and the lowest sensitivity can be found at 3% [95% CI: 1%-17%] from subjects with 0–7 days from symptom initiation, consistent with the biological time course of antibody production. During the highest infectious period (0–7 days from symptom initiation) the highest sensitivity was 21% [95% CI: 10%-38%], and during the declining phase of the infectious period (8–14 days from symptom initiation) the highest sensitivity was 61% [95% CI: 39%-80%].

Regarding the isotype target, the overall sensitivity was higher when incorporating either IgM or IgG detection (any positive isotype) for considering a test positive, followed by IgG only and then IgM only detection. As an example, if we look at the overall performance of the brand Access Bio, IgG or IgM detection had a sensitivity of 61% [95% CI: 51%-70%], and this was higher compared to detecting IgG only (57% [95% CI: 47%-66%]) or IgM only (29% [95% CI: 21%-39%]). Only when > = 15 days have elapsed since symptom onset and either IgM or IgG is present does the LFIA sensitivity goes over 90% for all four brands. When considering IgM only detection, the highest sensitivity amongst the four brands only reaches 54% [95% CI: 40%-67%]. A very high specificity (98% - 100%) was noted across all the brands evaluated.

## Discussion

In the present study, we evaluated the diagnostic performance of 4 different brands of LFIA COVID-19 IgG/IgM rapid test cassettes. The sensitivity of these tests was calculated using blood samples from RT-PCR confirmed positive patients for SARS-CoV-2. Even though all the tests reached a high specificity performance, none of them reached an overall high sensitivity. Only when assessing the sensitivity in patients with 15+ days from symptom onset that all brands perform optimally, having a sensitivity over 90%, being 94% [95% CI: 84%-98%] the highest sensitivity found amongst the four brands.

Serological immunoassays have variable diagnostic accuracy. According to the literature, the pooled sensitivity for serology tests (measuring for IgG or IgM) for CLIA, ELISA, and

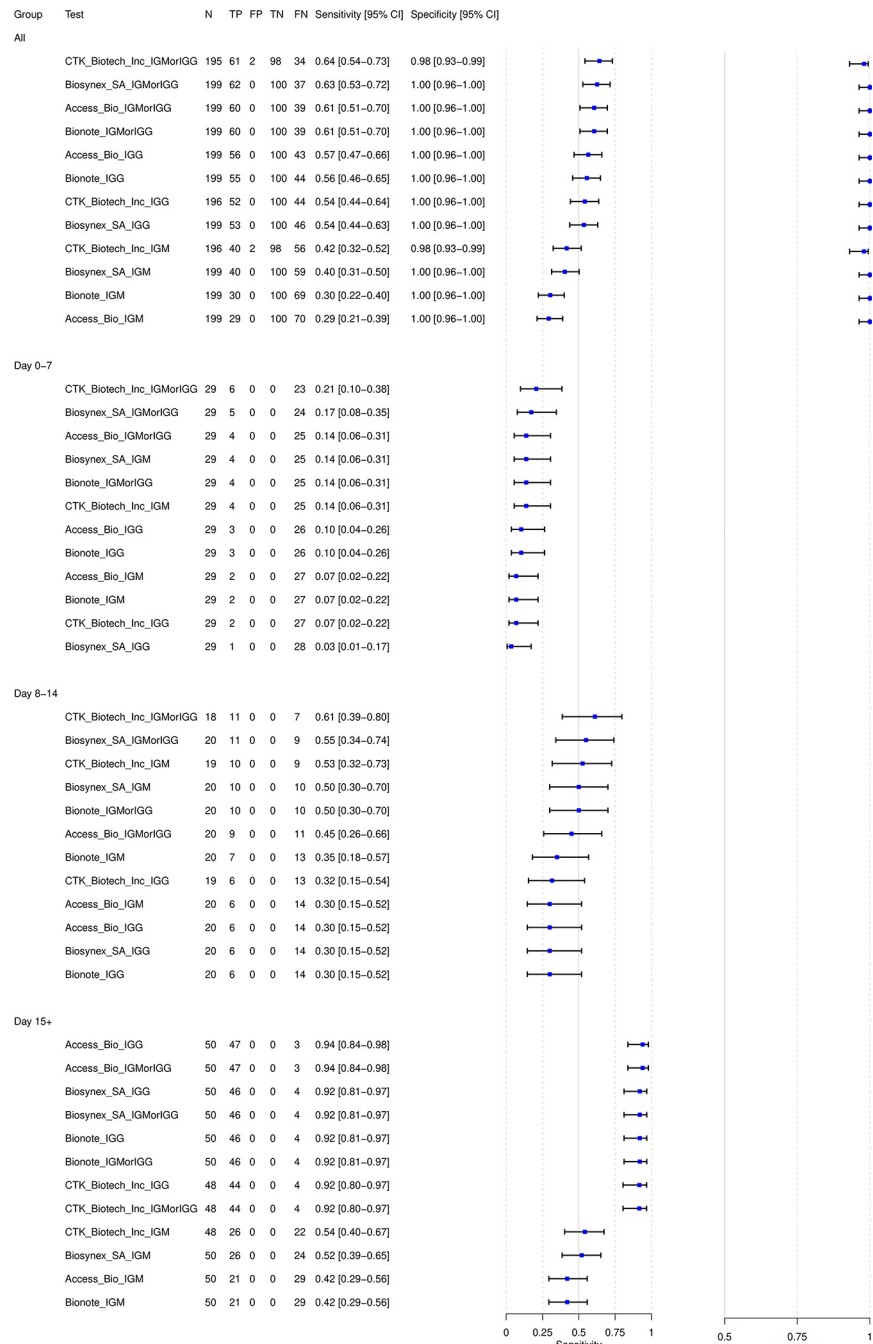

**Fig 1. Diagnostic accuracy of each antibody test and stratified by immunoglobulin antibody target and time since symptom onset.** TP: True positive, FP: False positive, TN: True negative, FN: False negative. IGG: detection of IgG line only; IGMorIGG: detection of IgG or IgM line; IGM: detection of IgM line only.

LFIA was 97.8% (46.2% to 100%), 84.3% (95% CI 75.6% to 90.9%), and 66.0% (49.3% to 79.3%), respectively; with pool specificities from 96.6% to 99.7% [10]. This has concordance with other meta-analyses, showing lower sensitivity for LFIA in comparison to CLIA and ELISA, and similar specificities between them [10, 36, 37]. It is worth mentioning that for the overall sensitivity of our brands for IgM or IgG detection, our tests results are similar to

previous meta-analyses sensitivity results (around 60%-75%) [37]; however, the sensitivity found in patients with 15+ days from symptom initiation in our brands were higher than 90%, whereas previous meta-analyses of LFIA tests have found this sensitivity to be somewhere between 70–80% [10]. The higher sensitivity for IgG only detection in our study compared to the IgM only detection is also in accordance with other studies performed worldwide [38, 39] and in our same region (South America) where other commercially available tests have also been evaluated [40].

There are multiple reasons why the antibody tests may have a lower capacity to recognize a COVID-19 positive case. The timing of the test is one of the most documented in the literature. Even though levels of IgM and IgG have been reported to be found since the day 4 after the beginning of the symptoms, elevated and more detectable levels could be reached by the second and third week after symptom onset [41]. IgM levels start to decline by the 4th week, and IgG levels remain way past the 7th week [41, 42]. This is also in accordance with our results where we found a low sensitivity for IgM and IgG detection in the first week, a higher sensitivity for detection of IgM in the second week, and an even higher sensitivity for detection of IgG after the 15th day. This is also supported by past reports with antibody test positivity happening in the intermediate or last phase of the disease [43].

Additionally, although we documented minimal false positivity (high specificity) in our study, there may be limitations to the included samples. For example, it has been previously documented that there is potential for cross-reactivity between dengue viruses and SARS-CoV-2. Up to 22% of pre-pandemic samples from patients with a previous diagnosis of dengue showed a positive result for SARS-CoV-2 antibody testing [44]. Given this estimate, caution should be exerted when using these antibody tests in tropical locations. As an example, a study that used LFIA antibody testing done in Peru, in the region of Iquitos (part of the Peruvian Amazonian jungle), showed a 71% SARS-CoV-2 seroprevalence [45]. Consequently, early on it was believed that Iquitos could be getting closer to reach herd immunity levels [45, 46]. However, this region is also known for having dengue as an endemic virus [47]; as such, this high seroprevalence might have been affected, to some degree by a potential cross-reactivity. A similar high estimated COVID-19 seroprevalence has also been reported in other Amazon jungle cities of Brazil (Manaus, 66%) [48] and Colombia (Leticia, 62%) [49], however, the latter used an antibody test validated using COVID-19 negative samples from patients infected with Dengue and other arboviruses.

The most described potential use for the LFIA and serological COVID-19 tests are as tools for serosurveillance studies. LFIA has been reported to have high specificity, similar to ELISA, but reduced sensitivity, which makes it ideal for large seroprevalence studies, helping in the understanding of the propagation of the pandemic and in the build-up of vaccination programs from LMICs. Additionally, depending on the viral antigen antibody identified by the test, there could be a potential for differentiating vaccination status, [24–26]; however, this should be used with caution depending on the days from symptom onset. Lastly, there are technologies evaluating the use of these LFIA antibody tests coupled with a spectrum-based reader (this analyzes the line color intensity) to provide quantitative results that previously found a high correlation with a surrogate neutralization test [31, 36], this could also be another venue for LFIA antibody testing use research.

## Conclusion

We evaluated the diagnostic performance of 4 different brands of LFIA COVID-19 IgG/IgM rapid test cassettes in samples from Peruvian population. Consistent with previous literature, we confirmed they had an overall high specificity and a variable sensitivity depending on the

days from symptom onset that was the highest after 14 days. All four immunological assays proved to be effective for identifying COVID-19 antibodies when two conditions were met: 1) at least 15 days have elapsed since symptom onset and 2) a sample is considered positive when either IgM or IgG is present. These LFIA antibody tests could be used for seroprevalence studies; however, attention should be put on the location where it would be employed and assess the potential for cross-reactivity with other endemic viral diseases. As well, caution with interpretation would be needed as these assays cannot distinguish between individuals who are seropositive due to prior infection or vaccination or both. Still, much information about immunity remains unknown. Future studies and LFIA point-of-care test manufacturers could aim to the identification of neutralizing antibodies, local accuracy considering endemic diseases, and its ability to detect and differentiate vaccinated and previously infected people, in order to find more and better uses for the serological COVID-19 tests.

## Author Contributions

**Conceptualization:** Rodrigo Calderon-Flores, Guillermo Caceres-Cardenas, Margaretha De Vos, Devy Emperador, Tatiana Cáceres, Luz Villa-Castillo, Audrey Albertini, Jilian A. Sacks, Cesar Ugarte-Gil.

**Data curation:** Rodrigo Calderon-Flores, Guillermo Caceres-Cardenas, Karla Alí, Margaretha De Vos, Devy Emperador, Tatiana Cáceres, Anika Eca, Luz Villa-Castillo, Audrey Albertini, Jilian A. Sacks, Cesar Ugarte-Gil.

**Formal analysis:** Rodrigo Calderon-Flores, Margaretha De Vos, Devy Emperador, Luz Villa-Castillo, Audrey Albertini, Jilian A. Sacks, Cesar Ugarte-Gil.

**Investigation:** Rodrigo Calderon-Flores, Guillermo Caceres-Cardenas, Karla Alí, Margaretha De Vos, Devy Emperador, Tatiana Cáceres, Anika Eca, Luz Villa-Castillo, Audrey Albertini, Jilian A. Sacks, Cesar Ugarte-Gil.

**Methodology:** Rodrigo Calderon-Flores, Guillermo Caceres-Cardenas, Margaretha De Vos, Devy Emperador, Tatiana Cáceres, Luz Villa-Castillo, Audrey Albertini, Jilian A. Sacks, Cesar Ugarte-Gil.

**Project administration:** Rodrigo Calderon-Flores, Guillermo Caceres-Cardenas, Karla Alí, Margaretha De Vos, Devy Emperador, Luz Villa-Castillo, Audrey Albertini, Jilian A. Sacks, Cesar Ugarte-Gil.

**Supervision:** Rodrigo Calderon-Flores, Guillermo Caceres-Cardenas, Karla Alí, Margaretha De Vos, Devy Emperador, Tatiana Cáceres, Luz Villa-Castillo, Audrey Albertini, Jilian A. Sacks, Cesar Ugarte-Gil.

**Validation:** Rodrigo Calderon-Flores, Guillermo Caceres-Cardenas, Karla Alí, Margaretha De Vos, Devy Emperador, Luz Villa-Castillo, Audrey Albertini, Jilian A. Sacks, Cesar Ugarte-Gil.

**Visualization:** Rodrigo Calderon-Flores, Margaretha De Vos.

**Writing – original draft:** Rodrigo Calderon-Flores, Guillermo Caceres-Cardenas, Karla Alí, Tatiana Cáceres, Anika Eca, Luz Villa-Castillo, Cesar Ugarte-Gil.

**Writing – review & editing:** Rodrigo Calderon-Flores, Guillermo Caceres-Cardenas, Margaretha De Vos, Devy Emperador, Tatiana Cáceres, Luz Villa-Castillo, Audrey Albertini, Jilian A. Sacks, Cesar Ugarte-Gil.

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
