## [Decision Letter · Decision Letter 0]

21 Feb 2023

PGPH-D-22-02126

Diagnostic performance of lateral flow immunoassays for COVID-19 antibodies in Peruvian population

Dear Dr. Ugarte-Gil,

Thank you for submitting your manuscript to PLOS Global Public Health. After careful consideration, we feel that it has merit but does not fully meet PLOS Global Public Health’s publication criteria as it currently stands. Therefore, we invite you to submit a revised version of the manuscript that addresses the points raised during the review process.

EDITOR: Please insert comments here and delete this placeholder text when finished. Be sure to:

Indicate which changes you require for acceptance versus which changes you recommendAddress any conflicts between the reviews so that it's clear which advice the authors should followProvide specific feedback from your evaluation of the manuscript

Please ensure that your decision is justified on PLOS Global Public Health’s publication criteria and not, for example, on novelty or perceived impact.

We look forward to receiving your revised manuscript.

Kind regards,

Miguel Angel Garcia-Bereguiain, PhD

Academic Editor

Journal Requirements:

Additional Editor Comments (if provided):

Dear Dr Ugarte-Gil,

After evaluation of your submission and the peer review reports, I believe your manuscript could be suitable for publication once the corrections and concerns made by the reviewers are addressed.

So, I invite you to submit a revised version of the manuscript. If you do not agree with some of the comments made by the reviewers, please explain it in your rebuttal letter.

Looking forward to receiving the new version of the manuscript.

Sincerely

Miguel Angel Garcia Bereguiain.

Reviewers' comments:

Reviewer's Responses to Questions

**Comments to the Author**

1. Does this manuscript meet PLOS Global Public Health’s publication criteria? Is the manuscript technically sound, and do the data support the conclusions? The manuscript must describe methodologically and ethically rigorous research with conclusions that are appropriately drawn based on the data presented.

Reviewer #1: Yes

Reviewer #2: Yes

2. Has the statistical analysis been performed appropriately and rigorously?

Reviewer #1: Yes

Reviewer #2: Yes

3. Have the authors made all data underlying the findings in their manuscript fully available (please refer to the Data Availability Statement at the start of the manuscript PDF file)?

Reviewer #1: Yes

Reviewer #2: Yes

4. Is the manuscript presented in an intelligible fashion and written in standard English?

Reviewer #1: Yes

Reviewer #2: Yes

5. Review Comments to the Author

Reviewer #1: The article entitled "Diagnostic performance of lateral flow immunoassays for COVID-19 antibodies in Peruvian population" has information of interest due to the evaluation of 4 brands that allow the serological diagnosis of antibodies in patients positive for SARS-CoV-2

Authors are requested to make the following corrections:

1. Correct the extra space in the word SARS-CoV-2 located in the Background section, add “and” in the numbering of study design cassette brands, add the acronym after Universidad Peruana Cayetano Heredia in Lima, Peru: UPCH.

2. Place in the abstract the brands

3. Consider the change in the name of the article, specifying the number of brands evaluated.

4. Table 1 can be deleted, just emphasize in the text that precedes it.

5. On page 6, add the references of the manuals corresponding to the evaluated brands.

6. Review the number of samples evaluated, there is inconsistency in the number both in the text and in Table 2-Figure 1 (Study samples and sample size and result sections).

7. I recommend that graph 1 become a table. Just specify the numeric data.

8. Among the studies carried out in South America, there is a study in Ecuador of interest that could be added to the discussion "Diagnostic Performance of Seven Commercial COVID-19 Serology Tests Available in South America"

These indications will allow clarity in the article. I believe that the article should be published to have more information on seroprevalence in countries where the variety of commercial brands does not allow knowing the efficiency of the tests.

Reviewer #2: Calderon-Flores and collaborators conducted a retrospective evaluation study of 4 LFIA antibody rapid tests for Covid-19. The study is well performed, the comments are below.

1. In the last paragraph of the background section it says:

“Our results contribute to the validation of these test brands to be used in seroprevalence studies and further evaluate its correlation with protective immunity or ability to detect vaccination/past infection status”

Overall, Covid-19 vaccines produce humoral immune responses to SARS-CoV-2 S protein, and antibodies to N protein appear due to infection, consequently, testing for protein-specific antibodies could be used to differentiate between different stages of the disease.

In this study, you don’t evaluate the correlation with protective immunity or the capacity of LFIA to detect vaccination or past infection status. Please, could you adjust this paragraph?

2. Could you clarify the first sentence of the last paragraph of the results section?

“Regarding the isotype target, the overall sensitivity was higher when incorporating either IgM or IG detection, followed by IgG and then IgM.”

3. Overall, the conclusions are a personal statement of your work and highlight the impact of the results, and future research that you considered needs to be done. So, What is the contribution of these references (32, 43–45) to the conclusions?

Maybe, it is more useful in the discussion section.

4. Reference number 17 should be properly formatted

6. PLOS authors have the option to publish the peer review history of their article (what does this mean?). If published, this will include your full peer review and any attached files.

**Do you want your identity to be public for this peer review?** For information about this choice, including consent withdrawal, please see our Privacy Policy.

Reviewer #1: **Yes: **MoralesD

Reviewer #2: **Yes: **Ismar A Rivera-Olivero

---

## [Editor Report · Decision Letter 1]

6 Apr 2023

Diagnostic performance of four lateral flow immunoassays for COVID-19 antibodies in Peruvian population

PGPH-D-22-02126R1

Dear Dr. Ugarte-Gil,

We are pleased to inform you that your manuscript 'Diagnostic performance of four lateral flow immunoassays for COVID-19 antibodies in Peruvian population' has been provisionally accepted for publication in PLOS Global Public Health.

Best regards,

Miguel Angel Garcia-Bereguiain, PhD

Academic Editor